# Microplastic-Enhanced Cadmium Toxicity: A Growing Threat to the Sea Grape, *Caulerpa lentillifera*

**DOI:** 10.3390/antiox13101268

**Published:** 2024-10-18

**Authors:** Weilong Zhou, Haolong Zheng, Yingyin Wu, Junyi Lin, Xiaofei Ma, Yixuan Xing, Huilong Ou, Hebert Ely Vasquez, Xing Zheng, Feng Yu, Zhifeng Gu

**Affiliations:** 1School of Marine Biology and Fisheries, Hainan University, Haikou 570228, China; weilong@hainanu.edu.cn (W.Z.); haolongzheng2022@hainanu.edu.cn (H.Z.); yingyinwu2023@163.com (Y.W.); 18760441727@163.com (J.L.); hlou@hainanu.edu.cn (H.O.); hebertely@163.com (H.E.V.); zhengxing_edu@163.com (X.Z.); 2Key Laboratory of Tropical Hydrobiology and Biotechnology of Hainan Province, Haikou 570228, China; 3Freshwater Fisheries Research Center of Chinese Academy of Fishery Sciences, Wuxi 214000, China; maxiaofei@ffrc.cn; 4Hainan Academy of Ocean and Fisheries Sciences, Haikou 570228, China; xingyixuan168@163.com; 5Sanya Institute of Breeding and Multiplication, Hainan University, Sanya 572024, China

**Keywords:** *Caulerpa lentillifera*, microplastics (MPs), cadmium (Cd), growth, physiological and biochemical analysis, transcriptomics

## Abstract

The escalating impact of human activities has led to the accumulation of microplastics (MPs) and heavy metals in marine environments, posing serious threats to marine ecosystems. As essential components of oceanic ecosystems, large seaweeds such as *Caulerpa lentillifera* play a crucial role in maintaining ecological balance. This study investigated the effects of MPs and cadmium (Cd) on the growth, physiology, biochemistry, and Cd accumulation in *C. lentillifera* while elucidating the underlying molecular regulatory mechanisms. The results demonstrated that exposure to MPs alone significantly promoted the growth. In contrast, exposure to Cd either alone or in combination with MPs significantly suppressed growth by reducing stem and stolon length, bud count, weight gain, and specific growth rates. Combined exposure to MPs and Cd exhibited the most pronounced inhibitory effect on growth. MPs had negligible impact while Cd exposure either alone or combined with MPs impaired antioxidant defenses and exacerbated oxidative damage; with combined exposure being the most detrimental. Analysis of Cd content revealed that MPs significantly increased Cd accumulation in algae intensifying its toxic effects. Gene expression analysis revealed that Cd exposure down-regulated key genes involved in photosynthesis, impairing both photosynthetic efficiency and energy conversion. The combined exposure of MPs and Cd further exacerbated these effects. In contrast, MPs alone activated the ribosome pathway, supporting ribosomal stability and protein synthesis. Additionally, both Cd exposure alone or in combination with MPs significantly reduced chlorophyll B and soluble sugar content, negatively impacting photosynthesis and nutrient accumulation. In summary, low concentrations of MPs promoted *C. lentillifera* growth, but the presence of Cd hindered it by disrupting photosynthesis and antioxidant mechanisms. Furthermore, the coexistence of MPs intensified the toxic effects of Cd. These findings enhance our understanding of how both MPs and Cd impact large seaweed ecosystems and provide crucial insights for assessing their ecological risks.

## 1. Introduction

The rapid pace of global industrialization has significantly contributed to marine environmental pollution, with microplastics (MPs) and heavy metals emerging as critical challenges in marine conservation [1,2]. MPs are defined as particles with a diameter of less than 5 mm and primarily originate from the degradation of plastic products such as polyethylene and polypropylene used in various human activities [3]. In recent years, these minuscule fragments have been detected across diverse marine environments, including coastal areas, open oceans, deep seas, and polar regions. They can be found in seawater, sediments, and even within marine organisms [4,5]. While plastics offer modern conveniences [6], their widespread presence as MPs has profound impacts on aquatic life, affecting plants, animals, and microorganisms alike [7]. Due to their small size, marine organisms easily ingest MPs, increasing their potential for harm [8]. Additionally, MPs can adsorb other pollutants such as heavy metals, persistent organic pollutants (POPs), and antibiotics, thereby compounding environmental risks [2].

Heavy metals, known for their toxicity and persistence in the environment, pose a significant threat to marine ecosystems. They accumulate in the environment through biological processes and undergo magnification along the food chain [9]. Among these metals, cadmium (Cd) emerges as an exceptionally hazardous pollutant due to its extensive industrial usage and high level of toxicity. Cd contamination in marine environments has numerous detrimental effects on aquatic life, potentially leading to mortality [10,11]. Cd not only undergoes migration and accumulation in aquatic environments but also exhibits bioaccumulation in marine organisms, thereby amplifying its presence through the food chain. This phenomenon leads to an elevated concentration of heavy metals in seafood, such as fish, which poses significant risks to human health [12]. Cd has been identified as a carcinogen, with ingestion linked to both cancerous and non-cancerous health issues in humans [13].

Large seaweeds play a pivotal role as primary producers within marine ecosystems by actively participating in carbon sequestration processes and contributing to carbon sinks [14]. Additionally, they serve as essential dietary components and valuable sources of biomaterials for industries such as pharmaceuticals, cosmetics, and hydrocolloids [15,16]. Seaweeds possess the capacity to absorb MPs and heavy metals from their surroundings, often becoming significant accumulation sites for these pollutants [17,18]. For instance, studies conducted by Li et al. [19] and Wickramasinghe et al. [20] have demonstrated that certain large algae in coastal areas accumulate MPs and various heavy metals. Interestingly, Znad et al. [21] suggest that this pollutant-capturing capability of large seaweeds renders them potential candidates for remediating polluted seawater. However, the negative impacts of MPs and heavy metals on large algae have been extensively documented. Research conducted by Li et al. [22] indicates that large seaweeds such as *C. lentillifera* and *Gracilaria lemaneiformis* not only accumulate MPs but also experience a reduction in photosynthetic oxygen evolution, an increase in malondialdehyde (MDA) content, and a decrease in extracellular polymeric substance (EPS) content. Similarly, Baumann et al. [23] discovered that high concentrations of metal ions negatively impact the growth and photosynthesis of marine macroalgae. Furthermore, Wang et al. [24] demonstrated that MPs alone or in combination with Cd can inhibit the growth of aquatic plants, potentially due to reduced chlorophyll content, impaired photosynthetic activity, and oxidative stress response.

*C. lentillifera*, commonly known as sea grape, consists of erect stems, stems, stolons, and filamentous pseudo-roots (Figure 1a). As a significant green alga, it serves as a fundamental component of the marine food web and holds significant economic value. Due to its abundant content of polyunsaturated fatty acids (PUFAs), essential amino acids, and low lipid content, it is frequently consumed as a salad and possesses considerable dietary value [25]. Additionally, its thalli are rich in polysaccharides and other bioactive compounds that offer antioxidant, antitumor, antiviral, and antidiabetic properties, thus rendering it highly valuable for pharmaceutical research and development [26]. *C. lentillifera* thrives in tropical and subtropical oceans, where MPs concentrations typically range between 2000 and 4000 particles/L, while levels of Cd^2^⁺, although usually low, can reach up to 500 μg/L during severe pollution events [27,28,29,30,31,32,33,34]. The ongoing degradation of the marine environment has adversely impacted *C. lentillifera*, compromising its food safety and economic value [35,36,37]. Furthermore, the accumulation of pollutants in this seaweed may pose potential risks to human health as they enter the food chain [38]. Despite these concerns, there is limited comprehension regarding the mechanisms through which these pollutants, particularly when evaluated in combination, impact on *C. lentillifera*.

Therefore, the primary objectives of this study are as follows: (1) to assess the impact of MPs on Cd uptake by *C. lentillifera*; (2) to investigate the effects of MPs and Cd, both individually and in combination, on the growth, tissue structure, and physiological and biochemical indices of *C. lentillifera*; (3) to elucidate alterations in antioxidant defense mechanisms under different stress conditions involving MPs and Cd; and (4) to verify the influence of MPs and Cd on the gene expression profiles of *C. lentillifera*. The findings from this study will enhance our understanding of the mechanisms behind toxicities caused by MPs and Cd, providing crucial evidences for ecological risk assessment, pollution control strategies, marine ecosystem protection, and human health preservation.

## 2. Materials and Methods

### 2.1. Experimental Materials

The *C. lentillifera* used in this study was collected from Changpo Town, Qionghai City, Hainan Province, China (110.61° E, 19.33° N). All thalli were derived from the same parent through asexual reproduction. Prior to the experiment, the thalli were thoroughly washed with sterile seawater. Stolons were cut to a length of 5.00 ± 0.43 cm from the tips using sterile dissecting scissors. These cut thalli were placed in cultivation tanks with artificially prepared seawater (Appendix A) for a 24 h recovery culture at 26 °C, under a light intensity of 500 lux.

### 2.2. Experimental Design

To simulate severe pollution conditions in tropical and subtropical seas, the experimental concentrations were set at 4000 particles/L for MPs and 500 μg/L for Cd^2^⁺ [30,31,32,33,34]. Fluorescent polystyrene microspheres with a diameter of 50 nm (Shanghai Zeyuan Biotechnology Co., Ltd., Shanghai, China) and analytical grade CdCl_2_ (Shanghai Aladdin Biochemical Technology Co., Ltd., Shanghai, China) were used to achieve these concentrations. The experiment consisted of four groups: the control group (without MPs or Cd^2^⁺), the MPs group (with the addition of MPs), the Cd group (with addition of Cd^2^⁺), and the MPs + Cd group (with co-addition of MPs and Cd^2^⁺). Each group was replicated three times. The cultivation was conducted in conical flasks with a volume of 1000 mL, each containing five thalli. The 15-day experiment was conducted in an incubator (GXM-358, Ningbo Jiangnan Instrument Factory, Ningbo, China) at a temperature of 26 °C under a light intensity of 3000 lux and a photoperiod cycle of 12 h dark/12 h light. The artificial seawater with respective concentrations of MPs and Cd^2^⁺ was changed every three days.

### 2.3. Growth Data and Morphological Characteristics Collection

The experiment commenced and concluded with the capture of photographs, as well as the measurement of growth indicators for each group. These indicators included stolon length, erect stem length, stolon branch number (the total number of five individuals per replicate), and weight. The lengths were measured using a vernier caliper (DL90150, Deli Group Co., Ltd., Ningbo, China), and thallus weight was determined with an analytical balance (ME104E, Mettler Toledo, Columbus, OH, USA). After the experiment, one thallus from each replicate was randomly selected for photography using a camera (T300, Sony Group Corporation, Tokyo, Japan). Weight gain rate (WGR, %) and specific growth rate (SGR, %) were calculated using the following equations:WGR = (W_2_ − W_1_)/W_1_ × 100,(1)
SGR = (Ln W_2_ − Ln W_1_)/t ×100,(2)
where W_1_ and W_2_ were the initial and final thallus weights, and t was the duration of the experiment.

### 2.4. Determination of Cd Concentration

The erect stem of *C. lentillifera* was digested in a polytetrafluoroethylene microwave tube, using a 0.1 g sample. The digested sample was diluted threefold with ultrapure water in a 50 mL centrifuge tube, agitated for 1 min, and filtered through a 0.22 μm membrane to remove impurities. The concentration of Cd^2^⁺ (μg/g) was measured using an inductively coupled plasma mass spectrometer (LabMS 3000, Lebertech, Beijing, China).

### 2.5. Determination of Antioxidant Capacity

At the conclusion of the experiment, healthy and intact thallus tissues were rapidly frozen in liquid nitrogen for antioxidant capacity analysis. For each replicate, 0.1 g of erect stem tissue was homogenized with a 0.9% sodium chloride solution at a ratio of 1:9, followed by centrifugation at 13,000× *g* for 10 min at 4 °C to separate the supernatant. After determining protein concentration using the BCA method, measurements of total antioxidant capacity (T-AOC), total superoxide dismutase (T-SOD), peroxidase (POD), catalase (CAT) activities, and malondialdehyde (MDA) content were measured using commercial kits (Nanjing Jiancheng Bioengineering Institute, Nanjing, China) according to the manufacturer’s instructions.

### 2.6. Transcriptome Analysis

At the end of the experiment, three thalli were randomly selected from each group for transcriptome analysis. Total RNA was extracted using TRIzol^®^ reagent, and its quality and concentration were assessed using a Bioanalyzer (5300, Agilent Technologies, Santa Clara, CA, USA) and a NanoDrop spectrophotometer (ND-2000, Thermo Fisher Scientific, Waltham, MA, USA). RNA purification, reverse transcription, library construction, and sequencing were performed by Majorbio Bio-Pharm Technology Co., Ltd. (Shanghai, China). Paired-end 150 bp sequencing was conducted on a NovaSeq 6000 sequencer. The obtained clean data were de novo assembled using Trinity, and functional annotation was performed using databases such as NCBI. Transcript expression levels were calculated using the TPM (Transcripts Per Million) method [39]. Differential expression analysis was carried out with the DESeq2 package (1.32.0) in R to identify genes that exhibited significant differential expression(|log_2_FC| ≧ 1 and *p* ≤ 0.05) which were referred to as differentially expressed genes (DEGs). GO and KEGG functional enrichment analyses of DEGs were performed in R.

### 2.7. Determination of Chlorophyll Content

After the experiment, three erect stem samples were randomly chosen from each replicate group to determine chlorophyll content. Chlorophyll was extracted using an anhydrous ethanol extraction method, and its optical density (OD) values were measured at 649 and 665 nm using a UV spectrophotometer (UV5, Mettler Toledo, USA). The absorbance values were then substituted into the following equations:Ca = 13.95A_665_ − 6.88A_649_(3)
Cb = 24.96A_649_ − 7.32A_665_(4)
to obtain the concentrations of chlorophyll A (Ca) and chlorophyll B (Cb) (mg/L). The total chlorophyll concentration was the sum of these values. Finally, the chlorophyll content was then calculated using the formula:Chlorophyll content (mg/g) = [Chlorophyll concentration × extraction volume × dilution factor]/fresh weight of sample(5)

### 2.8. Determination of Soluble Sugar Content

Upon completion of the experiment, three samples were randomly selected from each group for soluble sugar content determination. The 0.1 g sample was homogenized in a 1:9 ratio with distilled water. The resulting homogenate was transferred to a centrifuge tube and heated in a water bath for 10 min, cooled, and subjected to heating in a water bath for 10 min, followed by cooling and centrifugation at 4000 rpm for 10 min. Subsequently, the supernatant was diluted tenfold, and the determination of soluble sugar content was carried out according to the instructions (Nanjing Jiancheng Bioengineering Institute, Nanjing, China).

### 2.9. Data Processing and Analysis

The experimental data were presented as mean ± standard deviation. Statistical analysis was performed using DPS 14.5 software (Hangzhou Ruifeng Information Technology Co., Ltd., Hangzhou, China). Group comparisons were performed using one-way analysis of variance (ANOVA), followed by pairwise comparisons using the LSD test. Normality was assessed using the Kolmogorov–Smirnov test, and homogeneity of variance was checked using Cochran’s C test. Data that did not meet ANOVA assumptions were log-transformed, while percentage data were arcsine square root transformed. A significance level of *p* < 0.05 was considered statistically significant.

## 3. Results

### 3.1. Comparison of Growth Conditions

After 15 days of trial, notable differences in the morphology of thalli were observed among the experimental groups, as illustrated in Figure 1b. The control and MPs groups exhibited better growth compared to the Cd and MPs + Cd groups, with the latter displaying the poorest growth. Table 1 presents the statistical analysis of growth indicators. The erect stem length (FLE) and stolon length (FLS) followed this pattern: MPs > control > Cd > MPs + Cd. Specifically, the control group demonstrated the highest stolon branch number (NTB) at 13.00 ± 3.0, which was significantly greater than in the Cd group (6.33 ± 2.31) and MPs + Cd group (5.67 ± 3.79) (*p* < 0.05). However, no significant difference was observed between the control and MPs groups. Additionally, the MPs group recorded the highest final weight (FW) at 8.48 ± 1.11 g, significantly surpassing both the control and MPs + Cd groups. However, no significant difference was found between the MPs and Cd groups. The weight gain rate (WGR) was highest in the MPs group (119.10 ± 38.00%), significantly surpassing the rest of the groups (*p* < 0.05), with no significant difference between the Cd and MPs + Cd groups observed. The specific growth rate (SGR) was lowest in the MPs + Cd group (1.42 ± 1.82%/day), significantly lower than that of the highest observed in the MPs group (5.56 ± 1.33%/day) and the control group (3.93 ± 0.29%/day) (*p* < 0.05), while there was no statistically significant difference between the MPs + Cd and Cd groups.

### 3.2. Analysis of Cd^2^⁺ Content in Thalli

The Cd^2^⁺ content in thalli exhibited significant variations among groups, as shown in Figure 2a. The highest concentration of Cd^2^⁺ was observed in the MPs + Cd group (0.0316 ± 0.0010 μg/g), which significantly exceeded that of the other groups (*p* < 0.05). Thalli exposed to Cd alone also displayed significantly higher levels of Cd^2^⁺ content (0.0213 ± 0.0002 μg/g) compared to the control group (*p* < 0.05). The control group recorded the lowest Cd^2^⁺ content (0.0161 ± 0.0012 μg/g), with no significant difference from the MPs group (*p* > 0.05).

### 3.3. Comparison of Antioxidant Capacity

The results displayed in Figure 2b–f demonstrate the antioxidant capacity. The MPs group exhibited the highest POD activity (7.07 ± 4.15 U/mg protein), which was significantly higher than that of the MPs + Cd group (2.17 ± 0.65 U/mg protein), but not significantly different from the other groups (*p* > 0.05). In terms of T-SOD, the MPs group again showed the highest activity (62.34 ± 1.31 U/mg protein), which was significantly higher than all other experimental groups (*p* < 0.05), although it did not differ significantly from the control group either. The control group exhibited the lowest MDA content at 4.82 ± 1.94, which was significantly lower than both the Cd group (15.29 ± 1.01) and the MPs + Cd group (15.65 ± 6.74) (*p* < 0.05), but showed no significant difference compared to the MPs group. Additionally, the control group demonstrated the highest CAT and T-AOC activity, measuring 9.59 ± 0.53 U/mg protein and 10.71 ± 2.68 U/mg protein, respectively, which were significantly higher than those in both the Cd and MPs + Cd groups (*p* < 0.05), but not significantly different from the MPs group.

### 3.4. Sequencing and Gene Expression Analysis

After undergoing quality control, a total of 81.28 Gb of raw data was obtained, with each sample producing more than 6.34 Gb of clean data and Q30 exceeding 96.32% (Appendix A). The clean data were de novo assembled using Trinity, resulting in the generation of 50,065 unigenes and 66,820 transcripts with an average length of 853.95 bp and an N50 length of 1337 bp. Principal component analysis (PCA) revealed significant differences in gene expression levels between groups, with the Cd and MPs + Cd groups clustering together indicating similar expression patterns (Figure 3d). Differential expression gene (DEG) analysis showed that compared to the control group, the MPs group had a total of 384 DEGs (374 up-regulated and 10 down-regulated) (Figure 3a), the Cd group had a total of 3088 DEGs (328 up-regulated and 2760 down-regulated) (Figure 3b), and the MPs + Cd group had a total number of 2790 DEGs (182 up-regulated and 2608 down-regulated) (Figure 3c).

### 3.5. Functional Enrichment Analysis of DEGs

The GO enrichment analysis of DEGs (Figure 3e) revealed that the DEGs were primarily associated with cellular processes and metabolic processes in Biological Processes (BPs). In terms of Molecular Functions (MFs), DEGs were mainly involved in binding, catalytic activity, and structural molecule activity. Regarding Cellular Components (CCs), DEGs were predominantly distributed within cell parts, protein-containing complexes, organelles, organelle, and membrane parts. KEGG enrichment analysis revealed that DEGs in the MPs group were significantly enriched in the ribosome pathway (Figure 3f,j), whereas DEGs in the Cd group were primarily enriched in the photosynthesis pathway and photosynthesis–antenna protein synthesis pathway (Figure 3g,k). Similarly, DEGs in the MPs + Cd group were predominantly enriched in these photosynthesis-related pathways (Figure 3h,l). Notably, compared to the Cd group, DEGs in the MPs + Cd group were demonstrated to be significantly enriched in the ribosome, photosynthesis, and antenna protein synthesis pathways (Figure 3i). 

### 3.6. Chlorophyll Content

The chlorophyll content varied among the groups, as shown in Figure 4. There were no significant differences observed for chlorophyll A between the experimental and control groups (*p* > 0.05) (Figure 4a). However, the control group displayed significantly higher levels of chlorophyll B (0.115 ± 0.007 mg/g) compared to other groups (*p* < 0.05), with the MPs + Cd group exhibiting the lowest content (0.060 ± 0.002 mg/g). Moreover, there was no significant difference between the MPs + Cd and MPs groups (*p* > 0.05) (Figure 4b). The total chlorophyll content did not show a significant difference between the MPs and Cd groups compared to the control group at 0.225 ± 0.010 mg/g) but was significantly lower in the MPs + Cd group at 0.154 ± 0.010 mg/g (*p* < 0.05) (Figure 4c).

### 3.7. Soluble Sugar Content

The control group exhibited the highest soluble sugar content at 49.37 ± 12.74 mg/g, while the MPs + Cd group demonstrated the lowest content at 16.67 ± 10.01 mg/g. Although the addition of MPs alone did not significantly alter the soluble sugar content compared to the control group (*p* > 0.05), the addition of Cd or the combination of Cd + MPs significantly reduced the soluble sugar content in thalli (*p* < 0.05) (Figure 4d).

## 4. Discussion

Aligned with our initial hypothesis, the objective of this study was to assess the impact of MPs and Cd on *C. lentillifera*, both individually and in combination. The results provide insight into the distinct and synergistic effects of these pollutants on the physiological and biochemical responses of this essential marine algae.

### 4.1. Influence of Microplastics on Ribosomal Activity and Growth of C. lentillifera

The GO enrichment pathways (Figure 3e) indicated that MPs may impact the cellular structure and metabolic functions of *C. lentillifera* through mechanisms involving physical adsorption and chemical binding. In contrast, Cd appeared to primarily exert its effects by binding to intracellular proteins and enzymes, leading to oxidative stress and changes in gene expression patterns. The KEGG enrichment analysis revealed that the ribosome pathway was significantly affected by MPs, especially when combined with Cd exposure rather than Cd exposure alone. This suggests that MPs could have a substantial influence on protein expression levels, particularly those involving ribosomes which are complex organelles composed of ribosomal RNA molecules along with diverse proteins responsible for facilitating protein synthesis as well as regulating various biological processes [40]. As shown in Figure 3j, exposure to low concentrations of MPs resulted in the up-regulation of up to 72 ribosomal proteins (RPs). Ribosomes play a central role in cellular metabolism by translating messenger RNA (mRNA) into proteins during biosynthesis [41]. Moreover, ribosome biogenesis critically regulates cellular growth and proliferation [42]. In this study, the activation of pathways associated with ribosomes in response to MPs exposure likely enhanced the growth rate *C. lentillifera*, thereby maintaining ribosomal subunit homeostasis. These findings are consistent with the observed increase in specific growth rate among groups treated with MPs (Figure 3f,h,i). Similarly, Wang et al. [43] reported a significant height increment in peanuts treated with low concentrations of polystyrene (PS) and polylactic acid (PLA) microplastics, where MPs did not exhibit toxicity but instead promoted growth and nutrient uptake under certain circumstances.

Although the enhanced activity of the ribosome pathway was correlated with the overall weight gain rate and improved morphological indicators in *C. lentillifera* there was no significant difference observed in the specific growth rate between the MPs group (4.88 ± 1.33%/day) and the control group (4.28 ± 0.29%/day) (Table 1). This suggests that while low concentrations of MPs increased total biomass, they did not significantly accelerate the growth rate of the algae. Nonetheless, low concentrations of MPs still promoted algal growth, as evidenced by significant increases in the weight gain rate and stolon length. Furthermore, significant differences were observed between the MPs group and other experimental groups in terms of antioxidant markers such as T-AOC, T-SOD, POD, CAT, and MDA. However, no significant difference was found compared to the control group (Figure 2b–f). This indicates that while MPs significantly affected the ribosome pathway related to protein synthesis, they did not induce substantial oxidative stress within these algae. This could be attributed to the lower concentration of MPs used in this study compared to previous studies on rice, maize, and cabbage where concentrations of 50 mg/L, 20 mg/L, and 20 mg/L were employed, respectively [44,45,46]. The stability of soluble sugar content in algae further supports that MPs treatment did not disrupt energy metabolism. It is worth noting that although MPs treatment enhances biomass accumulation in *C. lentillifera* by modulating ribosomal protein expression, it has a limited overall impact on growth and physiological adaptation. These findings provide new insights into the ecological effects of MPs and emphasize the necessity for further exploration of their long-term ecological and physiological consequences.

### 4.2. Oxidative Stress and Photosynthetic Inhibition Induced by Cadmium Exposure

Cd is a highly toxic heavy metal that significantly impairs the growth of plants [47,48]. In this study, exposure to Cd notably suppressed the growth of *C. lentillifera*, as demonstrated by a significant reduction in branch number, stem length, weight gain rate, and specific growth rate compared to the control group. The adverse effects of Cd on *C. lentillifera* are primarily due to increased oxidative stress and disrupted photosynthesis. The schematic diagram of the effect of Cd^2+^ on *C. lentillifera* cells is shown in Figure 5, where exposure to Cd resulted in elevated levels of reactive oxygen species (ROS). This increase in ROS-activated pathways related to oxidative stress, such as the MAPK cascade, subsequently disrupts photosynthesis and other metabolic processes. Specifically, Cd exposure led to the down-regulation of key genes associated with Photosystem II (PSII) and Photosystem I (PSI), including PSBA, PSBR, and PSAA (Figure 3k). This down-regulation directly impaired the integrity and efficiency of the photosynthetic electron transport chain, thereby reducing the production of ATP and NADPH, which are essential products of the light reactions in photosynthesis. Consequently, this disruption in energy metabolism further enhanced the toxic impact of Cd on *C. lentillifera* [49]. Additionally, down-regulation of genes associated with the ATP synthase complex, such as *ATPF1D* and *ATPH*, also indicated impaired ATP synthesis. Photosynthesis serves as the primary driver of growth in plants and algae, with the down-regulation of photosynthetic genes shedding light on the molecular mechanisms underlying growth inhibition [50]. The decreased photoconversion efficiency limited the synthesis of photosynthetic products, thereby impacting carbon assimilation, which likely contributed to impaired morphological development and slower growth rates [51]. The observed reduction in soluble sugar content further indicated disruptions in carbohydrate metabolism, supporting the slowed growth observed (Figure 4d). Similar phenomena have been reported in studies investigating the effects of heavy metal stress on plants. Goyal et al. [52] demonstrated that Cd significantly affects photosynthesis and nitrogen metabolism enzyme activity in plants such as peas, barley, and alfalfa by reducing plasma membrane ATPase activity, altering membrane permeability, and disrupting metal ion homeostasis. These changes lead to stomatal closure, inhibited mineral absorption, disrupted water balance, slowed growth, reduced biomass, and potentially plant death. The impairment of photosynthesis, resulting in a disruption of the electron transport chain, leads to an excessive generation of electrons and energy, thereby forming highly reactive oxidants [53]. This inhibited the expression of antioxidant genes, such as those encoding POD and SOD, reducing the cell’s ability to scavenge reactive oxygen species (ROS) [54]. In this study, the decreased activities of T-AOC, T-SOD, POD, CAT, and significantly increased MDA content, as well as down-regulated expression of these enzyme-related genes, further support these findings (Figure 2b–g). The addition of Cd disrupted the balance of gene expression related to photosynthesis and the antioxidant defense mechanism, impeding energy synthesis and metabolic product formation, thereby leading to significant growth inhibition and morphological alterations in *C. lentillifera* (Figure 1b). The combined effects of impaired photosynthesis and reduced antioxidant capacity exacerbated oxidative stress in the plants, further compromising cellular physiological states [55,56]. These molecular-level changes ultimately manifested as growth inhibition and morphological alterations, revealing the complex and multi-level impacts of Cd stress on biological responses.

### 4.3. Synergistic Effects of Microplastics and Cadmium on Gene Expression and Cellular Function

As hypothesized, the combined exposure of MPs and Cd resulted in distinct alterations in gene expression, highlighting the intricate interplay between these pollutants in marine organisms [57]. In this study, we analyzed the combined effects of simultaneous exposure to MPs and Cd on *C. lentillifera* by exposing the algae to both contaminants simultaneously. The combined exposure group exhibited a suppressed number of budding structures, weight gain rate, and specific growth rate, along with a decrease in antioxidant enzyme activity and total antioxidant capacity, and an increase in oxidative damage. Transcriptomic enrichment analysis revealed that, in addition to photosynthesis, the combined exposure affected more crucial genes involved in the photosynthesis–antenna proteins pathway, thereby further limiting the efficiency of light energy capture and conversion. Notably, the changes observed in the ribosome pathway were a unique response compared to the addition of heavy metals alone (Figure 3i). The DEGs analysis showed that several key genes in the photosynthesis pathway, such as PSBO, PSBP, PSBQ, and PSAD, were consistently down-regulated in the combined exposure group compared to the control group (Figure 3l). This down-regulation may potentially compromise the functionality of photosystems and subsequently hinder the efficient conversion of light energy into chemical energy [58]. This finding is consistent with previous experiments solely focused on heavy metals, which also demonstrated inhibition of photosynthesis. Interestingly, exposure to both Cd and MPs resulted in a greater number of DEGs compared to Cd exposure alone (Figure 3c vs. Figure 3b). Specifically, the pathway related to photosynthesis–antenna proteins exhibited more pronounced down-regulation of DEGs including *LHCA3*, *LHCB1*, *LHCB2*, *LHCB4*, and *LHCB5* when exposed to both stressors as opposed to Cd exposure alone. These antenna proteins are essential components responsible for capturing and transmitting light energy within the photosystem [59]. Their down-regulation under combined stress conditions further restricts the efficiency of photosynthesis and energy capture. Studies suggest that a reduction in antenna proteins may also impair photoprotection mechanisms, making the algae more susceptible to light damage [60]. The diminished functionality of antenna proteins implies that *C. lentillifera* may face challenges in efficiently harnessing available light energy under light conditions, potentially placing it at a disadvantage in natural environments [61]. This reduced efficiency in capturing and converting energy could potentially lead to decreased biomass production, impacting primary productivity and food chain dynamics in aquatic systems. In comparison to Cd exposure alone, the presence of MPs exacerbated the toxic effects of Cd. Under the combined exposure, the growth performance of *C. lentillifera* was further inhibited, antioxidant enzyme activity decreased, and photosynthetic efficiency and energy conversion were disrupted. This may be due to the interaction between MPs and Cd, which could alter the bioavailability of Cd or the physical and chemical properties of MPs, enhancing Cd bioaccumulation and intensifying the toxic effects on the algae [62,63]. Additionally, MPs may physically obstruct or modify cell surface characteristics that affect nutrient absorption and metabolic processes, thus enhancing Cd bioaccumulation and exacerbating toxicity [64]. These findings are consistent with observations in terrestrial plants, such as maize and rice, where MPs treatment reduced growth and biomass accumulation, inhibited the efficiency of photosystem II (PSII), and increased levels of hydrogen peroxide (H_2_O_2_) and MDA, leading to oxidative stress [65,66]. In our study, the observed decrease in antioxidant enzyme activity, particularly the significant reduction in POD, along with a decline in photosynthetic pigments, especially chlorophyll B, suggests that under dual stress conditions compared to Cd alone, the state of cellular oxidative stress was further exacerbated. The inhibition of photosynthesis and the weakening of the photoprotection system under these conditions rendered the cells more vulnerable to ROS, resulting in additional damage to cell structure and function.

### 4.4. Ecological Risks and Industrial Implications of Combined Pollutants in Marine Environments

The results of this study demonstrate that the coexistence of MPs and Cd significantly exacerbates the negative effects on *C. lentillifera*. Although the current concentration of MPs in the South China Sea may not yet pose a significant threat to *C. lentillifera*, the coexistence of MPs and Cd led to a substantial increase in Cd concentration within the algae and more severe disruption of its physiological and biochemical functions (Figure 3a–f). Harmon et al. [67] highlighted that MPs impact aquatic organisms through physical and chemical pathways. The physical pathways primarily involve the mechanical inhibition of algal growth by MP particles, while the chemical pathways involve the toxicity of harmful chemicals carried by MPs, such as polycyclic aromatic hydrocarbons and plasticizers [68]. Although research on the synergistic effects of MPs and heavy metal compound pollution is relatively limited, this study specifically focused on investigating the combined effects of MPs and Cd. The findings revealed a significant increase in Cd accumulation in *C. lentillifera* due to MPs, which aligns with previous findings reported by Duan et al. [69]. This enhanced absorption mechanism may be related to the high activity of the MP surface and its ability to adsorb heavy metal ions [70]. Additionally, MPs have the potential to modify the properties of the algal surface, thereby increasing its affinity for heavy metals and exacerbating the bioavailability and toxicity of these metals.

This compound pollution poses significant risks to industries reliant on *C. lentillifera*, particularly the food and cosmetics sectors, which are highly sensitive to heavy metal content in raw materials [15,71]. While current pollution levels in the South China Sea are considered safe, the potential hazards arising from the synergistic effects of MPs and heavy metals could escalate with worsening pollution. This not only has implications for product safety but also has the potential to trigger a crisis of consumer trust, severely impacting the entire industry. Furthermore, *C. lentillifera* plays a crucial role in marine ecosystems by serving as a food source and habitat for various marine organisms [72].

The enhanced toxic effects of MPs and heavy metals can potentially impede the growth and reproductive capacity of these algae, thereby exerting an impact on the entire food chain and ecological balance. If polluted algae are consumed by organisms higher up in the food chain, toxins may accumulate, ultimately affecting organisms at higher trophic levels, including humans [73]. Therefore, despite current pollution levels not surpassing hazardous thresholds, it is imperative to maintain a high level of vigilance regarding the long-term consequences of compound pollution. Future research should further investigate the character of this compound pollution under varying environmental conditions, its specific impacts on marine ecosystems, and effective strategies for managing and mitigating these pollutants, emphasizing the importance of environmental protection efforts.

## 5. Conclusions

This study has revealed that low concentrations of MPs can promote the growth of *C. lentillifera* by activating the ribosome pathway. In contrast, Cd inhibits growth by suppressing photosynthesis, antioxidant responses, and energy regulation pathways, leading to reduced photosynthetic efficiency, diminished antioxidant capacity, and lower biomass accumulation. When *C. lentillifera* was co-exposed to both MPs and Cd, the presence of MPs enhanced Cd accumulation within the algae, severely disrupting photosynthesis, destabilizing the energy supply, and impairing cellular homeostasis, thereby exacerbating the toxic effects of Cd. These findings suggest that while MPs alone may promote growth to some extent, their combined exposure with Cd significantly amplifies the toxic effects, resulting in pronounced inhibition of algal growth and physiological functions. These results contribute to our understanding of the impacts of MPs and Cd, providing valuable data for further research on the implications of MPs and heavy metals in food safety and ecological risk assessments.

## Figures and Tables

**Figure 1 antioxidants-13-01268-f001:**
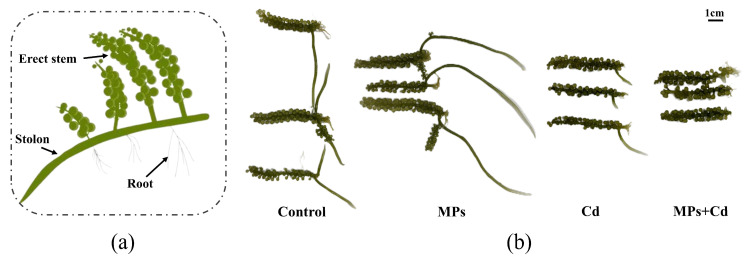
Growth and morphological changes of *C. lentillifera* after 15-day exposure to various conditions. (**a**) Schematic diagram of *C. lentillifera*. (**b**) Morphological comparison of *C. lentillifera* in the control, MPs, Cd, and MPs + Cd groups after a 15-day cultivation period.

**Figure 2 antioxidants-13-01268-f002:**
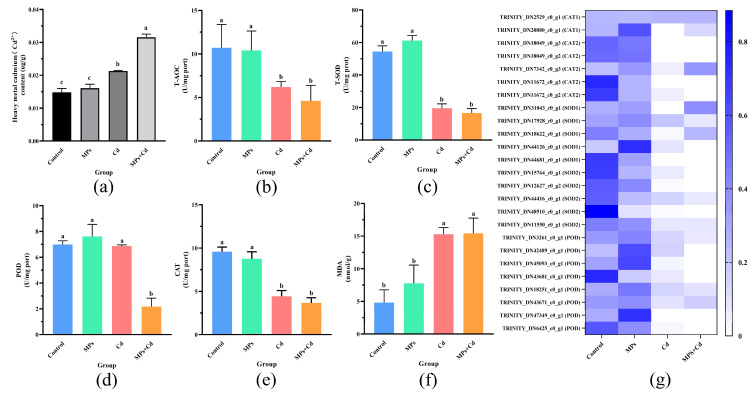
Analysis of cadmium accumulation, antioxidant capacity, and antioxidant gene expression in *C. lentillifera* under different treatment conditions. (**a**) Cd^2^⁺ content. (**b**) Total antioxidant capacity (T-AOC). (**c**) Superoxide dismutase (T-SOD) activity. (**d**) Peroxidase (POD) activity. (**e**) Catalase (CAT) activity. (**f**) Malondialdehyde (MDA) content. (**g**) Heatmap of antioxidant enzyme-related gene expression, based on normalized TPM data. The color intensity in the heatmap reflects the expression level of the gene, with the deeper blue indicating a higher level of expression. Different letters indicate significant differences between groups (*p* < 0.05).

**Figure 3 antioxidants-13-01268-f003:**
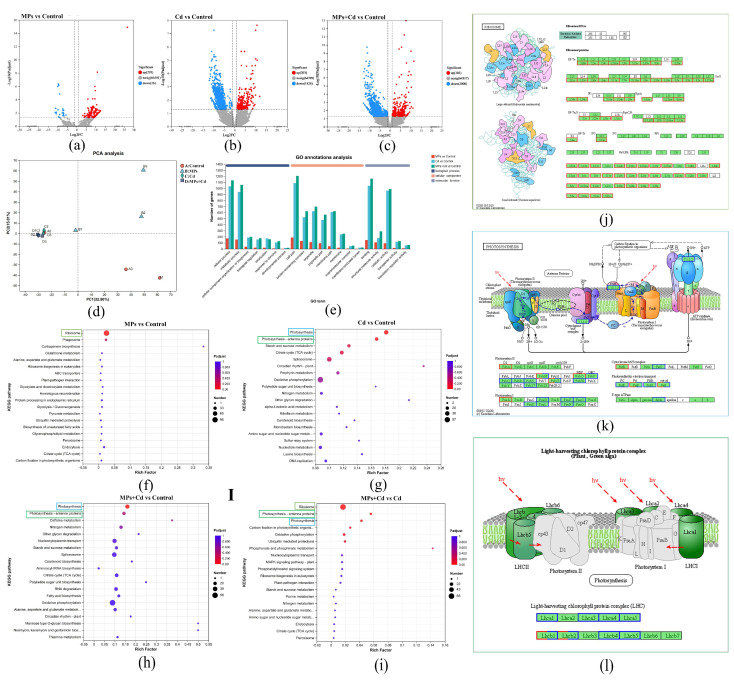
Differential gene expression and functional pathway enrichment in *C. lentillifera* exposed to microplastics and cadmium. (**a**) Volcano plot of DEGs between MPs and the control group. (**b**) Volcano plot of DEGs between the Cd and the control group. (**c**) Volcano plot of DEGs between the MPs + Cd and the control group. (**d**) PCA analysis among groups. (**e**) GO enrichment analysis of DEGs among groups. (**f**) KEGG enrichment analysis of DEGs between the MPs and the control group. (**g**) KEGG enrichment analysis of DEGs between the Cd and the control group. (**h**) KEGG enrichment analysis of DEGs between the MPs + Cd and the control group. (**i**) KEGG enrichment analysis of DEGs between the MPs + Cd and the Cd group. (**j**) Diagram of the ribosome pathway in MPs vs control. (**k**) Diagram of photosynthesis pathway in Cd vs control. (**l**) Diagram of photosynthesis-antenna proteins pathway in MPs + Cd vs control. In these diagrams, red boxes indicate genes with up-regulated expression, blue boxes indicate genes with down-regulated expression, green indicates annotated genes, and white indicates unannotated genes.

**Figure 4 antioxidants-13-01268-f004:**
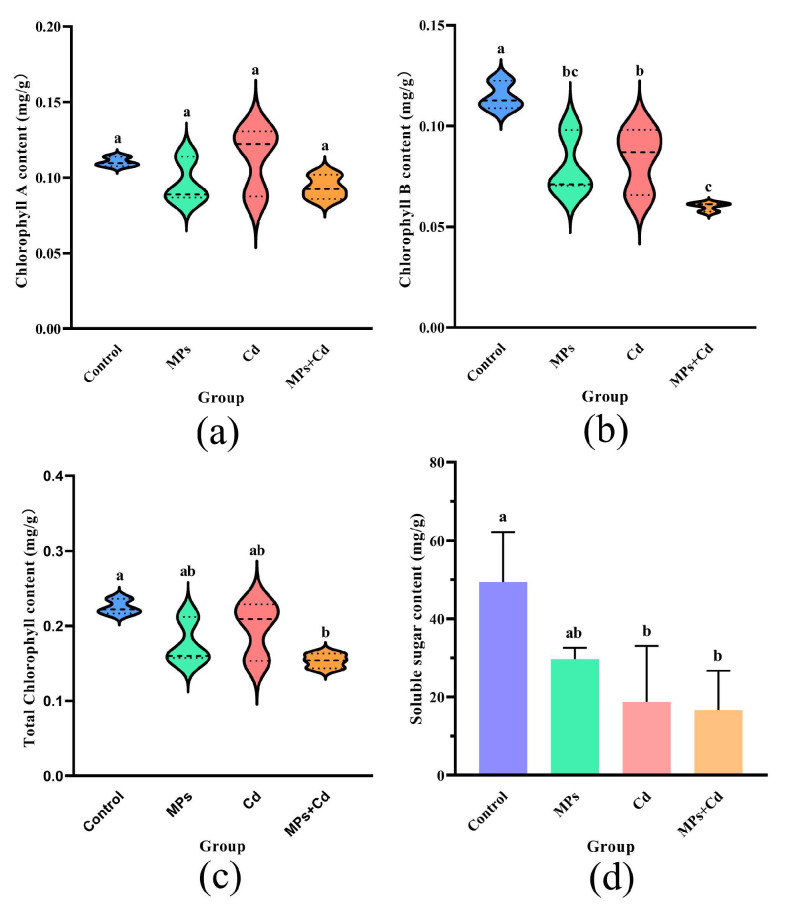
Comparison of the treatment groups in terms of chlorophyll and soluble sugar content in *C. lentillifera*. (**a**) Differences in chlorophyll A content. (**b**) Differences in chlorophyll B content. (**c**) Differences in total chlorophyll content. (**d**) Soluble sugar content in algal tissues among groups. Different letters indicate significant differences between groups (*p* < 0.05).

**Figure 5 antioxidants-13-01268-f005:**
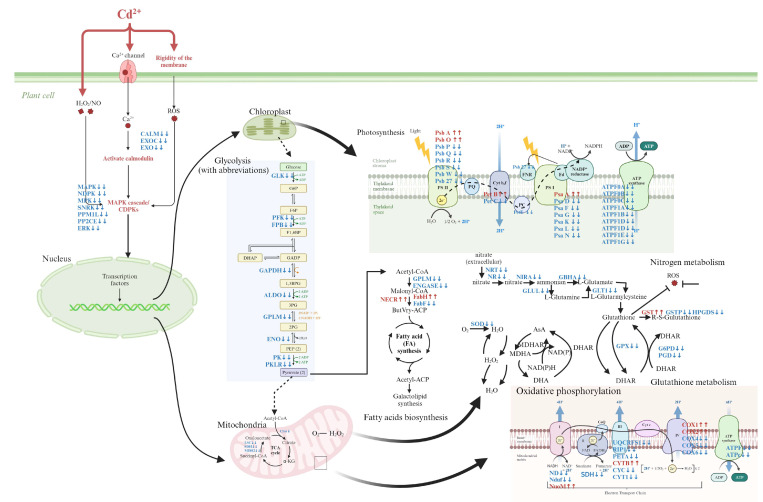
Schematic of the cellular response mechanisms of *C. lentillifera* to cadmium exposure. Changes in DEGs are indicated by colored arrows, with up-regulated and down-regulated DEGs shown in red and blue, respectively.

**Table 1 antioxidants-13-01268-t001:** Growth performance of *C. lentillifera* among different groups.

Growth Performance	Control	MPs	Cd	MPs + Cd
FLE (cm)	7.25 ± 0.30 ^b^	8.75 ± 0.62 ^a^	6.25 ± 0.15 ^c^	5.04 ± 0.47 ^d^
FLS (cm)	5.77 ± 1.81 ^b^	13.36 ± 1.04 ^a^	2.43 ± 1.06 ^c^	0.40 ± 0.17 ^c^
NTB (individual)	13.00 ± 3.00 ^a^	10.67 ± 2.08 ^ab^	6.33 ± 2.31 ^b^	5.67 ± 3.79 ^b^
IW (g)	3.47 ± 0.65 ^a^	3.87 ± 0.29 ^a^	3.49 ± 0.51 ^a^	3.71 ± 0.51 ^a^
FW (g)	5.95 ± 0.45 ^a^	8.48 ± 1.11 ^b^	4.35 ± 0.88 ^bc^	4.52 ± 0.55 ^c^
WGR (%)	74.14 ± 7.56 ^b^	119.10 ± 38.00 ^a^	24.23 ± 13.44 ^c^	24.53 ± 29.82 ^c^
SGR (%/day)	3.93 ± 0.29 ^a^	5.56 ± 1.33 ^a^	1.52 ± 0.77 ^b^	1.42 ± 1.82 ^b^

**Note**: Values were expressed as mean ± SD. Different letters indicate significant differences between groups (*p* < 0.05). **FLE**: Final length of erect stem; **FLS**: Final length of stolon. **NTB**: Number of stolon branches. **IW**: Initial weight. **FW**: Final weight. **WGR**: Weight Gain Rate. **SGR**: Specific Growth Rate.

## Data Availability

Data will be made available on request.

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
