# Peer review of "Microplastic-Enhanced Cadmium Toxicity: A Growing Threat to the Sea Grape, *Caulerpa lentillifera"

_antioxidants, 2024, doi:10.3390/antiox13101268_

Round 1

Reviewer 1 Report

This is a useful study that has used a range of endpoints to determine the response of Caulerpa to high levels of exposure to microplastics, cadmium and a combination of the two. The study includes molecular initiating events and a number of biochemical and morphological key events. It is well written but attention is needed for clarity of the figures (see detail comments).

Line 148: Stolon

Line191: add this on to the end of "following equations" rather than have as a separate sentence

Table 1: FLT - from the note below the table this should be FLE

NTB (individual)

Fig 2. it isn't clear to me what the heatmap colouring relates to. What is the axis showing? More explanation is needed in the figure legend. Why is (a) in greyscale and the rest in colour?

Fig 3. there is far too much information being presented in this figure and many of the details are virtually unreadable. I suggest you split this figure into three and amend the manuscript order and structure accordingly.

Fig 4. Why are there two visualization styles in evidence here? Either keep the violin plots or bars for all 4 panels but don't mix the two.

Author Response

Reviewer #1: This is a useful study that has used a range of endpoints to determine the response of Caulerpa to high levels of exposure to microplastics, cadmium and a combination of the two. The study includes molecular initiating events and a number of biochemical and morphological key events. It is well written but attention is needed for clarity of the figures (see detail comments).

Reply: We greatly appreciate your thorough review and valuable suggestions, which have undoubtedly improved the quality of our manuscript. We have carefully considered your comments and made the necessary revisions accordingly. Below are our detailed responses to each of your specific suggestions, and we hope the changes meet your expectations.

Specific comments

Comment 1: On Line 148 Stolon

Response 1: Thank you for your comment. I apologize, there was an error here, we accidentally duplicated the text, which has been removed from the manuscript. The corrected version is as follows (Lines 150-153):

These indicators included stolon length, erect stem length, stolon branch number (the total number of five individuals per replicate), and weight. The lengths were measured using a vernier caliper (DL90150, Deli Group Co., Ltd., China), and thallus weight was determined with an analytical balance (ME104E, Mettler Toledo, USA). After the experiment, one thallus from each replicate was randomly selected for photography using a camera (T300, Sony Group Corporation, Japan). Weight gain rate (WGR, %) and specific growth rate (SGR, %) were calculated using the following equations:

Comment 2: On Line 191 add this on to the end of "following equations" rather than have as a separate sentence

Response 2: Thank you for your comment. We have revised the sentence to add "following equations" at the appropriate position to improve readability, and this change can be found at Lines 151-152:

Weight gain rate (WGR, %) and specific growth rate (SGR, %) were calculated using the following equations:

Comment 3: On Table 1: FLT - from the note below the table this should be FLE

Response 3: Thank you for your correction. We have updated the table to change "FLT" to "FLE," and this correction can be found in Table 1 (Line 237) and its corresponding note. Additionally, we have also carefully revised the manuscript to ensure consistency, including updating "FLT" to "FLE" in Line 223:

The erect stem length (FLE) and stolon length (FLS) followed this pattern: MPs > control > Cd > MPs+Cd.

Comment 4: On NTB (individual)

Response 4: Thank you for your careful attention to detail. We have corrected the spelling of "NTB" as suggested, and the changes can be found in Table 1 (Line 237) of the revised manuscript.

NTB (individual)

Comment 5: On Fig 2. it isn't clear to me what the heatmap colouring relates to. What is the axis showing? More explanation is needed in the figure legend. Why is (a) in greyscale and the rest in colour?

Response 5: Thank you for your valuable comment. Sorry, here is where we did not explain clearly. The heatmap is based on the normalized TPM data of each group of antioxidant enzyme-related genes. The rightmost column indicates the correspondence between normalized gene expression levels and colors. The strength of the color reflects the expression level of the gene, with the deeper the blue indicating the higher the gene expression. We have added an explanation of this to the manuscript, as follows (Lines 259-262):

(g) Heatmap of antioxidant enzyme-related gene expression, based on normalized TPM data. The color intensity in the heatmap reflects the expression level of the gene, with the deeper blue indicating a higher level of expression.

Meanwhile, thank you for noticing the different colors in Figure 2a. The use of different colors was intentional. In Figure 2, the data in Figure 2a is of a different type of indicator than the other figures, so we used different colors to distinguish them. We believe that this design helps readers quickly identify different types of indicators so they can differentiate them.

Comment 6: On Fig 3. there is far too much information being presented in this figure and many of the details are virtually unreadable. I suggest you split this figure into three and amend the manuscript order and structure accordingly.

Response 6: Thank you for your suggestion. Indeed, as you feared, the information in Figure 3 is too much. When we initially designed Figure 3, we also tried splitting it into two or three parts. However, considering the completeness and continuity of transcriptome analysis, we preferred to keep the figure combined. At the same time, considering the magnifying feature of electronic journals, we maintained the current combined figure format. To solve the readability issue, we generated a higher resolution image to ensure that all details are clearly visible. We hope this will resolve the issue while preserving the structure of the article.

Comment 7: On Fig 4. Why are there two visualization styles in evidence here? Either keep the violin plots or bars for all 4 panels but don't mix the two.

Response 7:  Thank you for your comment. At first glance, it can be confusing to see two different visualization styles. However, we intentionally used two different visualization styles here for the same purpose as in Figure 2. In Figure 3, Chart D is a different type of metric than the others. Using different visualization styles can better distinguish different types of metrics, making it easier for readers to distinguish the data. If we changed them to the same style, it might reduce this distinction. Thank you for your understanding.

Reviewer 2 Report

The manuscript represents an important contribution to an evolving field.   The effect of increasing microplastics in the environment is serious, and is compounded by the presence presence of cadmium and other toxic metals.  The authors have added to the understanding of the molecular effects of both cadmium and microplastics separately and together on seaweed confirming their ecological risks.  The manuscript will provide an important addition to the literature in this field .  The presentation can be improved if the authors can improve the presentation of the Figures as suggested. 

Table 1 "FLT" should be enlarged

Author Response

Reviewer #2: The manuscript represents an important contribution to an evolving field. The effect of increasing microplastics in the environment is serious, and is compounded by the presence presence of cadmium and other toxic metals. The authors have added to the understanding of the molecular effects of both cadmium and microplastics separately and together on seaweed confirming their ecological risks. The manuscript will provide an important addition to the literature in this field. The presentation can be improved if the authors can improve the presentation of the Figures as suggested.

Reply: Thank you for your approval of our manuscript. We have carefully considered your comments and suggestions, and have further improved the clarity of the images. We hope that the revised manuscript will be satisfactory.

Comment 1: On Table 1 "FLT" should be enlarged

Response 1: Thank you for your comment. We have corrected the abbreviation "FLT" to "FLE" in Table 1 (Line 237) to ensure consistency, and the change can be found in the revised version.

Reviewer 3 Report

This work investigates the impact of micro plastics, cadmium, and their combination on the growth, tissue structure, and physiological and biochemical indices of Caulerpa lentillifera. It also examines alterations in its antioxidant defense mechanisms and the influence of microplastics and cadmium on its gene expression profiles.

The paper is well-structured, and all sections are clearly documented and distinct.
Only a few minor issues need to be addressed

3.1. Comparison of growth conditions, page 5, line 221. The authors should add in the manuscript that this number and std refer to NTB (as in Table 1).

Line 272-273: Since the authors added the total and average number of Gb for their data, I would also like to see the total number of reads for the same metrics and readers can read the Supplementary Table 2 for more information if needed.

Line 363: Delete the extra ”,” after (Table 1).

Figure 3: I would rather see this figure split in half. I am afraid there is a lot of information, particularly on the diagrams j, k and l, whereas details on various axes is lost as it is really hard to read.

Discussion, page 10, line 337. Please change alga to algae

4.1. Influence of microplastics on ribosomal activity and growth of C. lentillifera, page 10, line 349. Consider changing Fig 4j to 3j. 

Author Response

Reviewer #3: This work investigates the impact of micro plastics, cadmium, and their combination on the growth, tissue structure, and physiological and biochemical indices of Caulerpa lentillifera. It also examines alterations in its antioxidant defense mechanisms and the influence of microplastics and cadmium on its gene expression profiles. The paper is well-structured, and all sections are clearly documented and distinct. Only a few minor issues need to be addressed

Reply: Thank you very much for your approval of our manuscript. We appreciate your suggestion and have carefully revised the manuscript according to your feedback. Below are our responses to your specific comments.

Comment 1: On 3.1. Comparison of growth conditions, page 5, line 221. The authors should add in the manuscript that this number and std refer to NTB (as in Table 1).

Response 1: Thank you for your suggestion. We have clarified in the manuscript that the number and standard deviation refer to NTB, as indicated in Line 225.

Comment 2: On Line 272-273: Since the authors added the total and average number of Gb for their data, I would also like to see the total number of reads for the same metrics and readers can read the Supplementary Table 2 for more information if needed.

Response 2: Thank you for taking notice of our sequencing data. In total, the number of clean reads reached 542,356,722, with an average of 45,196,394 clean reads per individual. We have also added "Mapped reads" and "Mapped ratio" to Table 2 for your convenience, hoping that this will provide the additional information you are looking for. Thank you for your feedback.

Comment 3: On Line 363: Delete the extra ”,” after (Table 1).

Response 3: Thank you for your careful observation. We have deleted the extra comma after "Table 1" (Line 371).

Comment 4: On Figure 3: I would rather see this figure split in half. I am afraid there is a lot of information, particularly on the diagrams j, k and l, whereas details on various axes is lost as it is really hard to read.

Response 4: Thank you for your suggestion. Indeed, as you feared, the information in Figure 3 is too much, particularly on the diagrams j, k and l. When we initially designed Figure 3, we also tried splitting it into two or three parts. However, considering the completeness and continuity of transcriptome analysis, we preferred to keep the figure combined. At the same time, considering the magnifying feature of electronic journals, we maintained the current combined figure format. To solve the readability issue, we generated a higher resolution image to ensure that all details are clearly visible. We hope this will resolve the issue while preserving the structure of the article.

Comment 5: On Discussion, page 10, line 337. Please change alga to algae

Response 5: Thank you for your correction. We have changed "alga" to "algae" in the discussion section, as follows (Line 344):

The results provide insight into the distinct and synergistic effects of these pollutants on the physiological and biochemical responses of this essential marine algae.

Comment 6: On 4.1. Influence of microplastics on ribosomal activity and growth of C. lentillifera, page 10, line 349. Consider changing Fig 4j to 3j.

Response 6: Thank you for bringing this to our attention. We made a mistake here and have been revised as follows (Lines 356):

As shown in Figure 3j, exposure to low concentrations of MPs resulted in the up-regulation of up to 72 ribosomal proteins (RPs).